# A statistical fracture model for Antarctic glaciers

Veronika Emetc<sup>1</sup>, Paul Tregoning<sup>1</sup>, and Malcolm Sambridge<sup>1</sup> <sup>1</sup>Research School of Earth Science, Australian National University, Canberra, Australia *Correspondence to:* Veronika Emetc (Veronika.Emetc@anu.edu.au)

## Abstract.

Antarctic and Greenland hold more than 99% of all fresh water on Earth and, therefore, can significantly influence global sea level. Predicting future ice sheet mass balance depends upon ice sheet modelling, but it is limited by knowledge of a number of processes, some of which are still poorly understood. One such process is the calving of the ice shelves, where blocks of

- ice break off from the ice front. However, large scale ice flow models do not include an accurate representation of this process and the most commonly used damage mechanics and fracture mechanics methods have a large number of uncertainties. Here we present an alternative, statistics-based method to model the most probable zones of nucleation of fractures. We test our theory on all main ice shelf regions in Antarctica, including the Antarctic Peninsula. We can model up to 99% of observed fractures, with an average rate of 77% which represents a 50% improvement over previously used damage-based approaches,
- thus providing the basis for modelling calving of ice shelves. We found that classifying Antarctic ice shelf regions based on the factors that controlled fracture formation led to grouping of ice shelves/glaciers with similar physical characteristics and geometry.

Keywords. Antarctica, glaciers, probability, calving, ice shelves, fracture nucleation, crevasse, logistic regression, bayesian

## 1 Introduction

- In recent years, increased positive temperature anomalies have been observed in Antarctica (Jansen et al., 2007; Vaughan et al., 2003; Johanson and Fu, 2007; Steig et al., 2009). Future climate changes in this area may be even more pronounced (Vaughan et al., 2003), which may cause the state of the Antarctic ice sheet to change significantly. This could lead to a release of fresh water currently stored in the ice sheet and a consequent rise in sea level. West Antarctica alone can contribute up to 4.3 metres to global sea level (Fretwell et al., 2013). Almost 30% of the global population lives in coastal areas (Small and Nicholls,
- 2003), therefore, the effect of sea level rise (SLR) can significantly impact the economy and society in these regions. However, projections of global sea level rise have a number of uncertanties, with the greatest related to the calving rate of the ice shelves (Jansen et al., 2007). Thus, understanding the factors that control the mass balance of the Antarctic ice sheet is crucial if we want to better understand the future impact of climate change and contribution of Antarctic ice mass loss to global SLR.

The mechanics of ice mass loss of Antarctica is controlled by three processes: surface ablation, basal melting and calving (Paterson, 2000), where the later relates directly to SLR for grounded ice and contributes to buttressing effect on floating ice shelves. Due to the fact that the surface ablation in Antarctica is relatively low, it was previously believed that the Antarctic ice

sheet was stable. However, increased calving from the major ice shelves between 1998 and 2003 led to a growing concern of the total ice sheet stability (Borstad et al., 2013). It has been shown that the main part of ice mass loss from Antarctica comes from an increased number of iceberg calving events (Mercer, 1978; Jacobs et al., 1992; Katz and Worster, 2010; Gudmundsson, 2013; Borstad et al., 2013) and estimates of the calving effect on the mass balance are as high as 50% (Depoorter et al., 2013; Rignot et al., 2013). Also, studies by Jezek (1984); De Angelis and Skvarca (2003); Dupont and Alley (2005); Goldberg

5 Rignot et al., 2013). Also, studies by Jezek (1984); De Angelis and Skvarca (2003); Dupont and Alley (2005); Goldberg et al. (2009); Katz and Worster (2010); Gudmundsson (2013); Borstad et al. (2013) showed that increased calving can lead to destabilization of ice shelves and thus to a loss of the supporting mechanism they provide on the inland ice in Antarctica. This support can be crucial for the overall stability of the ice sheet (Miles et al., 2013).

However, none of the IPCC projections of the future state of Antarctica considers the calving mechanisms (Stocker, 2014).
Research on crevasse propagation started as early as 1955 and calving parametrisation has being under development for the last 20 years. However, a method that can universally describe calving at any ice shelf in Antarctica has not been found. In most of the large scale ice sheet models this process is still either described with a large number of limitations or not implemented at all (e.g. Benn et al. (2007b); Stocker (2014); Bassis (2011)). There are a number of calving parameterisations, but still none of them includes propagation of fractures in both depth and latitude-longitude space. Also, most of the available models are

- specific to a particular case and set of predictors, or a location, and therefore can not be applied to any chosen ice shelf. Many experiments have tried to find an optimal way of implementing a full calving process to ice sheet models, but none of them has worked for all locations in Antarctica. However, a number of different approaches based on Linear Fracture Mechanics (LEFM), Continuum Damage Mechanics (CDM) or "crevasse-depth" have advanced the understanding of the calving process. The history of development of these methods is presented in Table 1.
- ELMER/Ice and PISM models only include calving built on simplified physics (Alley et al., 2008). PISM can perform a calving algorithm based on eigen vector theory introduced by Levermann et al. (2012) (Eq. 1) (Winkelmann et al., 2011), which is only a first order approximation and does not include initiation and propagation of crevasses. They calculate calving rate as:

$$C = Kdet(\dot{\epsilon}) = K\dot{\epsilon}_{+}\dot{\epsilon}_{-}, \text{ for } \dot{\epsilon}_{\pm} > 0, \tag{1}$$

where K is a proportionality constant and  $\dot{\epsilon}_{\pm}$  are the eigenvalues of the horizontal strain rate tensor.

Another method they use is calculating a calving rate based on the critical ice thickness, which is mainly used to model calving of marine terminated glaciers rather than floating ice shelves (due to different physics governing calving between the grounded ice and floating ice). Also, most of the experiments with ELMER/Ice calving were performed for Greenland glaciers, which have a different calving mechanism to the floating ice shelves in Antarctica (Van der Veen, 2002).

In Table 1 we can see that to date Continuum Damage Mechanics (Kachanov, 1958) and Linear Fracture Mechanics (Van der Veen, 1998a) are the most common methods to model fracture formation. Also, Krug et al. (2014) showed that combining these two methods provides a significant improvement towards modelling of calving.

5

In order to develop a reliable calving law we need to have a strong basis that would include both physics and observational data. In other words, if we want to model propagation of fractures and the consequent calving it is essential to know the location of these fractures on the ice shelf surface or where they are initiated. In this study, we focus on modelling of crevasses (surface fractures less than 200 metres wide) on the surface of the Antarctica ice sheet. We implement a method based on a probabilistic approach that can be used as an alternative to the damage-based approach mentioned above. Our method is build on the logistic regression algorithm (LRA) and includes finding a relationship between the observed fractures we take from satellite images and various factors such as geometry, mechanical properties, flow regime as well as climate forcing (that we

use as predictors). Our best results were achieved combining LRA with Bayesian as well as Jensen-Shannon Divergence theory

described in section 3. We apply our method to most of the ice shelf regions in both West and East Antarctica as well as the
Antarctic Peninsula. From modelling of 34 ice shelves/glaciers we found that we can match the observations of fractures with a success rate from 18% to 99% with an average rate of 77% (Figure 1a).

# 2 Background

# 2.1 Current state of calving computations in ice sheet models

Damage mechanics is used to calculate zones where ice is weakened due to formation of small fractures. However, this method
depends on a number of control predictors that have been calculated only for one specific glacier in Greenland (Duddu and Waisman, 2012) and therefore can lead to incorrect results when applied to a randomly selected antarctic ice shelf/glacier. Furthermore, in situ observations in Antarctica are not feasible due to a very large regional extent of the ice shelves as well as very high frequency of fracturing (sometimes every 50 metres). Also, satellite observations alone are difficult to use because of the very high resolution required to see all the small fractures and snow can often cover the fractures in the image. Therefore, the main factors that creates weaknesses in ice shelves/glaciers in Antarctica remain unclear.

The Linear Fracture Mechanics approach is focused on calculating the stress intensity around fractures and their propagation until the stress intensity reaches a certain critical value. Although the "crevasse-depth" criterion based on the shear rate estimation with depth can produce interesting results when applied to marine terminated glaciers in Greenland (Nick et al., 2010), it can not describe calving at the floating ice shelves in Antarctica due to the different mechanics. The Continuum Damage

- 25 Mechanics approach, based on the method suggested by Kachanov (1958), includes estimation of damaged zones where the ice is softened due to presence of fractures. Damage can be calculated from the inversion of velocities at the ice shelves (Borstad et al., 2012) which is based on minimising the cost function that describes the misfit between the observed and modelled velocities. It is proposed that it can allow to model zones where the ice is weakened and thus we can estimate where the initiation of small fractures takes place. Krug et al. (2014) used damage mechanics to model the slow development of small fractures in ice.
- 30 Consequently, the estimation of the depth of these initial fractures is performed using the stress intensity calculations, which is based on Linear Fracture Mechanics. It can describes propagation of the formed fractures with depth and a consequent calving that occurs mainly at the ice shelf front.

The ELMER/Ice model (Gagliardini et al., 2013) includes calving combining CDM and LEFM, but this method was performed for Greenland only (Krug et al., 2014) and has not been proven to work for any ice shelf in Antarctica. The Community Ice Sheet Model (CISM) assumes that calving takes place when the water depth reaches a certain value (Price et al., 2014), which is, as mentioned above, suitable only for tidewater glaciers, but not for ice shelves. The Parallel Ice Sheet Model (PISM)

5 have an implemented calving parametrisation, but it is based on simplified physics and includes only along-flow ice spreading (Alley et al., 2008). They suggest two options for calving. The first is based on the assumption that calving takes place when thickness at the ice calving front reaches a critical ice thickness H<sub>cr</sub> ~ 150 - 250m. The second applies the Eigen vector principal components proposed by Levermann et al. (2012), which is based on the correlation between the large-scale calving rate and the local ice-flow spreading rate. However, it considers only large-scale behaviour and does not take into account the 10 formation and propagation of crevasses.

We present the main studies of calving in Table 1 and among them the most comprehensive parametrisation of calving by Krug et al. (2014) includes combining damage and fracture mechanics. This method is more complex in comparison to the other approaches and can help to understand calving mechanism in Antarctica much better. Apparently, the propagation of fractures can be modelled only if the fractured zones are known or computed in a good agreement with the observed surface

fractures. Therefore, modelling of the formation of the fractured zones is an important basis for the consequent estimation of the fracture depth as well as calving and it has to be described in the ice sheet models in an accurate way.

Damage is an arbitrary variable used to determine failure of ice and nucleation of fractures (usually when the damage predictor reaches a certain value) and can be described as a density function. However, the damage approach might not be able to provide us with reliable results that agree with observations, because the nature of fracturing of the ice sheet might be

- too complex to be comprehensively described by the methods of damage mechanics. Therefore, damage may not be justified for glacier ice, especially at the ice front (Levermann et al., 2012). The propagation of damage is usually calculated using an advection equation and a source function. However, this method has a number of uncertainties. First of all, the damage is not a physical predictor and modelling its propagation includes many complications when choosing an appropriate source function and when including a healing process. In fact, the source function is the controlling factor in the damage propagation, but the
- hypothesis of what should be used as the source function has not been well justified. The one that is used in the models can be constructed in either a complex or simplified way, but always includes many assumptions, arbitrary predictors and uncertainties (Pralong and Funk, 2005; Krug et al., 2014). Secondly, the estimation of the main predictors that determine damage is based on experiments that have a large number of uncertainties (Krug et al., 2014). It also does not take into account such important factors as ice fabric, impurities and non-linear behaviour due to presence of crevasses (Borstad et al., 2012). Also, it is sensitive
- to a number of constants such as initial damage  $D_0$  (Borstad et al., 2012), especially when applied to a randomly chosen ice shelf/glacier. Thirdly, when calculating damage from an inversion in Ice Sheet System Model (ISSM) (Larour et al., 2012) (described in the next section) it is impossible to do an inversion on both the basal friction and damage since they have similar effects on the ice velocity (Morlighem, 2017). Therefore, the inversion for damage is performed only on floating ice, where there is no friction, and, thus, there is no information about the damage on grounded ice.

# 2.2 Ice Sheet System Model ISSM. Model setup

First, we used ISSM (Larour et al., 2012) to compute factors such as velocities, stresses, strains, backstresses, the dynamics of the ice sheet in time as well as friction coefficient and viscosity (calculated from inverse modelling) in order to use them in our statistical model as quasi-observations. All our experiments were performed for 34 regions in Antarctica (see Figure 1b), each including both ice shelves and the grounded ice around 100 kilometres from the grounding line (further referred as ice shelf

5 including both ice shelves and the grounded ice around 100 kilometres from the grounding line (further referred as ice shelf regions or ice shelf/glacier). Second, we calculated damage on floating ice for each region from inversion of velocities (Borstad et al., 2013). This method finds a value of damage from an inversion based on minimising the misfit between observed and modelled velocities.

ISSM is a fully dynamic model that includes both 2-D and 3-D components and is based on the equations of conservation of mass, momentum and energy as well as Glen's law (shows the relation between shear strain rate and shear stress). There are a number of approximations to solve these equations that are available in ISSM: Shallow Ice Approximation (SIA) is generally used for grounded ice and assumes that the height-to-width ratio is small, Shelfy Stream Approximation (SSA) and Pattyn's higher order model (HO that can be applied for a 3D case). We modelled glacier dynamics in a two-dimensional case in order to achieve faster computational time while having a relatively high spatial resolution and apply the SSA approximation (assumes

that the vertical shear is zero, thus velocities do not depend on depth (MacAyeal, 1989)) as it is more suitable for modelling velocities at the floating ice shelves in a 2-D case.

To perform a simulation ISSM requires forcing data, geometry of the region, boundary conditions as well as friction and rheology coefficient. We used seaRISE air temperature, snow accumulation and geothermal heat flux (Le Brocq et al., 2010) as climate forcing data. Also, we calculated changes of the surface temperature with the lapse rate and imposed it on the ice

- surface. The data for geometry of the ice shelves and surrounded grounded ice (such as bedrock topography, ice thickness and glacier surface) were taken from Bedmap2 at 1 km spatial resolution bedrock topography. We used them for both: as an input for modelling with ISSM and the predictor factors in our probabilistic method. Basal friction for grounded ice and rheology for floating ice were calculated from inversion of velocities (Khazendar et al., 2007). The horizontal ice velocities for the inverse modelling were taken from InSAR (450-metres resolution) (Rignot et al., 2011b, a) and we applied Dirichlet conditions at the
- inflow boundary. The ice mask to distinguish between grounded and floating ice is based on hydrostatic equilibrium (mask  $\geq 0$  if ice is present and mask  $\leq 0$  for the ocean). We set boundary conditions as follows: the upper surface is considered stress-free and at the ice-bedrock interface the friction is applied. The mean basal melting rate for grounded and floating ice (Depoorter et al., 2013) were set as 2 and 4 m/yr, respectively, and the thickening rate at the base of the floating ice (due to ice accretion) was set to 2 m/yr (a default setting in ISSM). It is a first approximation, but in this study we do not model the basal effect on
- horizontal pattern of fracturing at the surface. We ran one simulation for stress balance solution per region (ice shelf/glacier) which allowed us to obtain the factors required as an input in the calculation of the probability of fracturing.

In addition, it is important to have a finer resolution mesh in order to model surface fractures, as the distance between them is normally around 50-100 metres. However, using a mesh of 50-metre or 100-metre resolution creates a significant increase of the computational time in our model. Therefore, for our experiments, we constructed a 200-metre resolution triangular mesh

for the chosen domains in East and West Antarctica as well as the Antarctic Peninsula. First, all the main predictors were calculated on a 200-metre resolution mesh (to achieve a faster computational speed) and then the results were interpolated to a 100-metre resolution mesh (to use in our fracture model resolved at 100-metres). All further computations and analysis were performed on this finer mesh.

# 5 3 Methods

To develop an alternative method for modelling fracture formation in ice shelves/glaciers, we took into account that the damage varies from 0 to 1, in a probabilistic sense with 0 being not fractured and 1 being fully fractured. We can substitute this probability with a likelihood function. Thus, to implement our ice calving into the ice sheet models, we developed a statistical-based parametrisation for fracture formation in ice shelves/glaciers. First, we present the main method (logistic regression) used

10 for predicting formation of fractures. Second, we describe the predictor factors (predictors) we used in our analysis. Finally, two methods used for optimisation of a set of predictor factors are presented (Bayesian based algorithm and Jensen-Shannon Divergence).

# 3.1 The logistic regression algorithm (LRA)

The function P is called a logistic function. Taking any range of data it produces values from 0 to 1 and thus it can be used
as a probability (Hosmer Jr and Lemeshow, 2004). The logistic function allows us to calculate the probability of an event as a function of different predictor factors (see Table 2).

Our set of predictors  $x_i$  initially consisted of 24 vector arrays. Some of them are known to influence fracture formation (stresses, strains, changes of the velocity gradient, ice rheology). Others we considered to be potentially important (various geometrical properties, proximity to the ice front and the grounding line, etc). We analysed the data for each predictors and found that there were a linear dependancy between first and second components of the principal stress axis. Also, temperature and accumulation were excluded from the list of predictors due to the incompatibility of their spatial resolution with the relatively fine 100-metre mesh we used to model fractures. However, they might be important for the formation and propagation of fractures as warmer temperature can increase the number of fractures, but a better resolution climate data would be needed. Excluding this two factors, we obtained a set of 20 predictors that we describe further in this section.

To apply the logistic regression algorithm we constructed a function P (Eq. 2) that describes the probability of fracture nucleation in a certain node as a function of the predictor factors  $x_i$ . Each j element in the P array can be calculated as:

$$P_{j} = Prob(X = 1|x_{i}) = \frac{\exp(\beta_{0} + \beta_{1} \cdot x_{1j} + \beta_{2} \cdot x_{2j} + \beta_{3} \cdot x_{3j} + ...)}{1 + \exp(\beta_{0} + \beta_{1} \cdot x_{1j} + \beta_{2} \cdot x_{2j} + \beta_{3} \cdot x_{3j} + ...)},$$
(2)

where for  $x_{ij}$  is an element in a predictor array (*i* is the number of the predictor and *j* is a row number). The unknown coefficients  $\beta_i$  can be found by maximising the likelihood function *L* (Eq. 3).

$$L(\beta_j) = \prod_{j=1}^n P_j^{\delta_j} (1 - P_j)^{1 - \delta_j},$$
(3)

where n is the number of observations and  $\delta$  is the Kronecker symbol:

$$\delta = \begin{cases} 1, & \text{when there is a surface fracture visible on a satellite image} \\ 0, & \text{otherwise.} \end{cases}$$
(4)

All the predictor factors are either taken from InSAR and Bedmap2 or calculated using formulas described below. The calculation of each predictor was performed using methods already implemented in ISSM (e.g. stresses, strains, friction coefficient) or

by applying new algorithms (e.g. calculation of curvature, distances to ice front, grounding line, mountains). First, the effective strain rate  $\dot{\epsilon}_E$  had to be included in our analysis because it is known that a crevasse initiation is linked to strain rates (Campbell et al., 2013). We calculated this predictor in ISSM as:

$$\dot{\epsilon}_E = \sqrt{\frac{\epsilon_{xx}^2 + \epsilon_{yy}^2}{2} + \epsilon_{xy}^2},\tag{5}$$

where  $\epsilon_{ik} = \frac{1}{2} \left( \frac{\partial v_i}{\partial x_k} + \frac{\partial v_k}{\partial x_i} \right)$  and  $v_i$  are horizontal components of the ice flow velocities (InSAR). Also, we included principal strains, calculated as:

$$\mu = \frac{\epsilon_{xx} + \epsilon_{yy}}{2} \pm \sqrt{\left(\frac{\epsilon_{xx} - \epsilon_{yy}}{2}\right)^2 + \epsilon_{xy}^2},\tag{6}$$

Second, principal stress  $\lambda$  is important as it has a direct effect on the opening and closing of crevasses. Also, principal axis s 20 shows the direction of the stresses (compressive or tensile):

 $\sigma \cdot \mathbf{s} = \lambda \mathbf{s} \tag{7}$ 

Thirdly, when ice is flowing over a vertical bend on the surface it might experience fracturing, therefore we included the surface and bedrock slopes calculated from:

$$\begin{cases} \frac{dS}{dx} = \sum_{i=0}^{M} \alpha_i S_i \\ \frac{dS}{dy} = \sum_{i=0}^{M} \gamma_i S_i \\ \text{Slope} = \left(\frac{dS}{dx}\right)^2 + \left(\frac{dS}{dy}\right)^2 \end{cases}$$
(8)

where *M* is the number of vertices, *S* is the surface,  $\alpha_i = \frac{d\phi_i}{dx}$  and  $\gamma_i = \frac{d\phi_i}{dy}$  are the coefficients of a nodal function defined 5 as  $\phi_i = \sum_{i=1}^{3} \alpha_i \cdot x + \gamma_i \cdot y$  (calculated for each triangular element on the mesh, the constant term is equal to 0).

Fourth, curvature of the glacier channel can play a role in fracture formation as we can observe more fractures around a horizontal bend of the glacier. The calculation of this predictor was performed using:

$$\alpha = \arccos\left(\frac{v_x(P) \cdot v_x(E) + v_y(P) \cdot v_y(E)}{|\boldsymbol{v}(P)| \cdot |\boldsymbol{v}(E)|}\right),\tag{9}$$

- where v(P) is the ice velocity at the point of observations and v(E) is the velocity D metres away from the point. The distance D is based on the velocity magnitude v(P), because if the velocity is high we need to increase D so that two subsequent points capture the geometry of the bend. Thus, if v(P) is greater than 2000 m/yr we assign D = 3v(P), otherwise D = 6v(P).
- By looking at the satellite images we can see that more fractures occur at a certain distance from the ice front as well as the grounding line. We found that the relation between the occurrence of fractures and distance to the ice front as well as the distance to the grounding line is non-linear. For most ice shelves/glaciers we can see more fractures 3-5 km as well as 10-13 km away from the front and a slightly smaller number of fractures closer than 3 km to the front or between 5 and 10 km. Therefore, instead of using d<sub>IF</sub> and d<sub>GL</sub> (distance to the ice front and the grounding line in km) as predictor variables, we construct dummy variables: DM<sub>IF</sub> and DM<sub>GL</sub>, respectively, which represent two-column arrays in the following form:

$$\mathbf{DM_{IF}} = \begin{cases} (1,1), & \text{when } 3\text{km} \le \mathbf{d_{IF}} 

Lastly, because all the predictor factors have different units as well as significantly different magnitudes we normalised all variables, using:

 $x_i^* = \frac{x_i - \mu_i}{\sigma_i}$ , where  $\mu_i$  and  $\sigma_i$  are mean and standard deviation of the predictor variables, respectively. (12)

# 5 3.2 Observing fractures using satellite images

In order to obtain information about the location of fractures on the ice sheet surface we used satellite images taken from Google Earth-Pro, where images of the Antarctic ice sheet were available at different spatial resolutions. However, to be able to see small surface fractures, our choice was limited only to the images with a resolution smaller than 10 metres for the period between 2011 and 2015. Also, we included only regions with a good satellite coverage (at least one hight resolution satellite image) and where it is relatively easy to identify surface fractures.

Many features can be observed on the ice surface and it is important to distinguish the surface fractures from other patterns such as surface troughs due to bottom crevasses or subglacial channels. It was suggested by Luckman et al. (2012) that the features on the images that are wider and have a larger spacing between them are more likely to be troughs linked to bottom crevasses. Also, Alley et al. (2016) proposed a way to distinguish basal channels and fractures on the satellite images. They

classified channels into sub-glacially sourced, ocean-sourced and grounding-line-sourced. Most of them either follow the ice flow direction and begin either at the grounding-line or downstream. Therefore, we avoided including such features in the set of chosen surface fractures.

In addition, it is important to choose satellite images of different regions in Antarctica in order to build correct predictions for fracturing  $P_i$  as the diversity in sampling provides a better estimation of the  $\beta$  coefficients (the number of observation points is less important). Thus, by choosing multiple glaciers we can more accurately construct an approximate surface that separates

fractured from non-fractured nodes (the plane is determined by  $\beta$  coefficients).

To construct the set of observed fractures we manually selected fractured points as well as non-fractured points on the satellite images. Non-fractured points are more difficult to identify, therefore we mainly take them from areas where high-resolution images are available and they are mostly located in blue ice regions. These are the areas with low snow accumulation or where

the snow was removed by the wind. In such areas we can clearly see where the ice is not damaged. We construct a field of fractures by assigning fracture nodes to 1, non-fractured nodes to 0. We select fractured nodes where we can see surface fractures that do not have features cause by present basal fractures or channels.

The resolution is not high enough in all areas to clearly see every fracture and also some of them can be covered in snow. This creates a large uncertainty in cells where we are not able to observe any visible fractures. Sometimes it is impossible to

10

20

say whether there are no fractures or whether we just cannot see them. Thus, we corrected a field of fractures by using the probability of observed fractures instead of the observations of fractures in order to make the observation field continuous as well as to account for the uncertainties in our ability to observe accurately whether or not areas are fractured. Firstly, we assumed that if there is a fracture in one node then the neighbour nodes are more likely to be fractured as well. In the node

where we could not see a fracture we assigned the observed probability to 1. Within a radius of 500 metres away from the fracture we decreased the probability linearly from 1 to 0.55. If a non-fractured node was found within a high resolution area we assigned the probability of fracturing in this node to 0.05. Within a 500 metres radius of non-fractured nodes we allowed the probability to increase linearly from 0.05 to 0.4. In all other nodes we set a value of 0.5, assuming that, since the information in the satellite image is ambiguous, it is an equal probability of being fractured or non-fractured.

# 3.3 Optimisation problem

## 3.3.1 Random walk

To implement our method we used a 100-metre resolution models for 34 ice shelf regions (including floating ice and surrounded grounded ice). We started with an apriori model of fracturing and then improved it based on three methods: random walk, Bayesian and Jensen-Shannon Divergence algorithm. The main difficulty in constructing the probability function for each

- Bayesian and Jensen-Shannon Divergence algorithm. The main difficulty in constructing the probability function for each glacier was to identify a set of the predictor factors we need to include in LRA. As the logistic regression formula was based on the correlation coefficients between the observed fracture field and a certain set of 20 predictors in our model, we needed to determine which predictors we need to include in the formula. For each glacier we performed a 100000-step run with random sets of predictors for each step (number of predictors and the selection of predictors are selected at random every
- step). We defined the best-fitting model to be a result with a success of identifying fractures more than 70% and the error of over-estimation smaller than 15% (however, if a good-fit model was not found after 2000 steps we looked for a model with a 65% success and 20% error). Once a good fit was found, we saved it and continued running the model with different sets of predictor factors for the remaining number of steps to see if a better model can be found. This also provided a mean set of factors needed for a good-fit model.

# 20 3.3.2 Bayesian

To test the behaviour of the models with more precision and to choose an optimal set of factors from the full set we also performed a non-linear Bayesian inversion. This process has the advantage of allowing us to take into account uncertainties in the observed data.

We assume that the prior PDF is uniformly distributed between 1 and 20 (U[1,20], because the maximum number of predictor factors in a set is equal to 20). As a prior model we take a calculated probability at every time step. Each step included two criteria: if a new likelihood is greater than the prior likelihood or it is greater than a certain percentage (taken at random at each step) of the old likelihood we accept the model. However, the main difficulty arose when we tried to find the expression for a likelihood function for our model. We tested a number of different commonly used expressions, such as:

$$\begin{cases} L_i = \sum \log(f_i) \\ f_i = (1 - p_i)^{1 - d_i} + p_i^{d_i}, \end{cases}$$
(13)

$$\begin{cases} L_i &= \sum \log(f_i) \\ f_i &= (1 - p_i) \cdot (1 - d_i) + p_i \cdot d_i, \end{cases}$$
(14)

where  $d_i$  and  $p_i$  are observed and modelled fractures on a glacier, respectively.

However, all of them produced very large likelihoods that increased dramatically with a percentage change in probability 5 density function, reaching an order of  $10^5$ . Thus, it was important to choose a representative likelihood function. We constructed the likelihood function assuming that the measure R of the total agreement between two models (the sum of all probabilities) follows a Gaussian distribution with a mean E (Eq. 15) and a standard deviation as a square root of variance  $\sigma$  described in Equation 18.

We calculated the expected values for both the data and a chosen model as:

10 
$$E(f_i^{pred}) = \sum_{i=1}^N f_i$$
 (15)

$$E(f_i^{obs}) = \sum_{i=1}^{N} d_i^2 + (1 - d_i)^2$$
(16)

$$E(f_i^{best}) = \sum_{i=1}^{N} p^* \cdot d_i + (1 - p^*) \cdot (1 - d_i), \tag{17}$$

where  $p^*$  is the best-fit probability and the probability  $f_i$  that two predictions agree in a cell *i* is calculated using Equation 14. Second, we found the variance as a difference between the two expected values (Eq. 18).

$$\sigma = |E(f_i^{obs}) - E(f_i^{best})| \tag{18}$$

Our idea was to calculate the likelihood  $L_i$  as an exponential function of the misfit between data and the model assuming that either data (observed fractures) or the analysed model have an error (Eq. 19). Where misfit  $\phi_d(m)$  can be calculated as the square of a ratio between the expectation for the data minus the expectation for the model and the variance of the data.

$$L_{i} = e^{-\frac{1}{2}\phi_{d}(m)}, \text{ where } \phi_{d}(m) = \frac{(E(f_{i}^{obs}) - E(f_{d}^{pred}))^{2}}{\sigma^{2}}.$$
(19)

In addition, the area of each ice shelf region is an order of  $10^8 km^2$  which leads to a very large sum of all modelled probabilities between 0 and 0.5 and therefore an extremely large likelihood (it is important to notice that this values should not be

confused with 0 and 1 values set for observed fractures only). In order to achieve a more realistic magnitude of the likelihood function we needed to re-calculate the estimated probabilities by scaling them between 0.55 and 1. To do this we assigned everything below 0.55 to non-fractures (zero values) and scaled the remaining values to the range 0 to 1.

For a prior model and prior scores, we took the best-fitting model from the random walk search described above. First, we
perform a Bayesian analysis for 500 steps and after we narrowed down the selection and we accepted only those models that have likelihoods greater than 90% of the best likelihood.

#### 3.4 Glaciers classification and Jensen-Shannon Divergence (JSD)

We started with a construction of a binary array for each glacier, where the number of rows represent the number of good-fitting models for a glacier and the number of columns represent the 20 predictor factors. Next, we found the average occurrence of

10 each predictor:

$$A_i = \frac{\sum_{j=1}^N k_j}{N},\tag{20}$$

where  $i \in [1, 20]$  is the predictor index, N is the number of good-fitting models and  $k_j = 1$  when the predictor is included in the good-fit model j and 0 otherwise.

This allowed us to find how often a certain predictor was included in good-fit models. If a predictor was selected more than 50% of the time then it was assigned as a potential for a best-fit mode, otherwise it is set to 0. Thus, we obtained a 34x20 array (34 glaciers vs. 20 predictors) that consisted of 1 when the predictor can be included in the best-fit model and 0 otherwise.

Next, we classified the glaciers in 4 groups. There were a large number of possible combinations to select such groups. Therefore, we constructed a test that can run through every possible combination and calculate the percentage of a similarity
between glaciers in a group (Eq. 21).

$$S = \frac{M}{21} \cdot 100,\tag{21}$$

where M is the number of matches between sets of predictors for two glaciers and S is a group number.

Thus, we placed all 34 glaciers in 4 different groups with Group 1 having glaciers that can be more easily combined and 25 Group 4 being a more narrow group of specific glaciers that can not be placed in any of the other three groups.

In order to validate of our method we applied the additional Jensen-Shannon Divergence method (JSD) to identify the similarity between the best probability for each glacier and a probability calculated by placing the glacier in a certain group.