# Peer review of "A statistical fracture model for Antarctic glaciers"

_The Cryosphere, 2017_

## Referee Comment (RC1) · Anonymous Referee #1 · 23 Jun 2017

"The formation of fractures is a very complex process that cannot be effectively described by only applying damage mechanics."

"Previous analysis based on damage accounts for stresses, thickness and viscosity, but ignores such predictors as proximity to mountains and grounding line as well as the curvature of a channel, which might be crucial for modelling of the fracture formation in Antarctica."

This is not serious text - nothing breaks because it is close to mountains or a grounding line. Such parameters may be correlated to fracture only if they correlate indirect via ice mechanics. This is like correlating drowning accidents to ice cream eating. It is rather easy to construct such measures, but there is nothing to be learned from them. The connection are superficial.

For this paper to be published a more relevant crevasse formation measure should be constructed. All included parameters should at least be thoroughly motivated.

---

## Referee Comment (RC2) · Anonymous Referee #2 · 25 Jul 2017

This manuscript describes a new probabilistic method for representing the location of fractures in Antarctic ice shelves (although the title indicates "glaciers" it is really ice shelves that are the focus). Rather than focus on the physics of ice fracturing, the proposed approach lumps as many observational factors as possible into a probabilistic framework which then searches for a best-fit combination of factors that produces a probabilistic field (analogous to damage) that agrees with observations of fractures individually picked from satellite images. The method is compared to a selective use of continuum damage mechanics, and the authors claim that their new approach is much better.

The introduction and background is rather meandering, and paints a quite critical picture of the use of fracture mechanics and damage mechanics for representing fractures

in glaciers. Indeed, the naive reader might be left with the impression that these methods are completely arbitrary and without merit. It does not appear that the authors have a very thorough awareness and understanding of the damage mechanics literature as it has been applied in recent years to glacier and ice shelves. Inversions for damage of the type presented in the manuscript only produce damage in areas where fractures are actively forming or widening. It is important to distinguish between the "high-advection lifecycle" and "low-advection lifecycle" crevasse definitions of Colgan et al. (2016). If crevasses have advected far from where they formed, then the appearance of a crevasse is not a representation of local stresses/strain rates! In the presented statistical method, no difference is made between where fractures are initiating versus where they have been advected for long distances after initiation. This history-dependence is very important, and is a key shortcoming of the present approach. It also makes the comparison with the damage inversion a sort of apples-vs-oranges comparison. The two approaches shouldn't necessarily produce the same thing.

As for the statistical approach, there are many arbitrary and strange choices in its formulation. It seems as if every possible observational factor has been thrown into the mix just to see what comes out. Surprisingly, factors like the principal stress axes (components of unit normal vectors) and proximity to mountains end up being predictive, even if they have no physical relation to fracture mechanics, which should be the foundation of even a statistics-based fracture model. In the end, the most heavily influential factors in the statistical model are factors that would lead to higher stresses (and thus higher damage or predictions of crevasse depth) in a properly formulated and initialized model. This would seem to actually argue in favor of continuing with physical models such as the continuum damage mechanics models that are roundly criticized in the manuscript.

The manuscript would benefit from a careful rewrite to avoid the use of vague or unscientific language, and to better describe the background material and theory. The modeling methods and results need to be described and shown in much more detail in

order for a competent peer to be able to attempt to reproduce the study and check its findings.

Modeling description

The description here of the use of damage mechanics is lacking in sufficient detail both to judge and to be able to reproduce the study. The authors are very vague in describing how they calculated damage for comparing to their statistical results. I am suspicious of the damage results, although there is simply not enough information for me to say one way or another whether their results are reasonable. I am not convinced that the new statistical model is actually better, in any quantifiable sense, than a properly formulated damage model.

The description of the modeling using ISSM is incomplete and vague. You mention that you use ISSM "to compute factors such as velocities, stresses, strains, backstresses, the dynamics of the ice sheet in time as well as friction coefficient and viscosity (calculated from inverse modelling)..." This language ("factors such as...") is too vague for such a methods description. Describe specifically what you calculated and how you calculated it. Calculating "the dynamics of the ice sheet in time" could mean anything from modeling one year of an ice shelf to modeling a paleo-reconstruction of an ice sheet. Did you really model transient ice sheet dynamics?

More information is needed on how the model geometry is initialized. How was the grounding line determined? Did you use a mask other than that provided by Bedmap2 for defining floating/grounded portions? This is important, as Bedmap2 is missing many ice rises and ice rumples that are very important for ice shelf dynamics and particularly for stress calculations (see for example Furst et al., 2015; Matsuoka et al., 2015).

In order to invert for damage, you must have a reasonable idea of the ice temperature in order to parameterize the flow rate factor ('B' in Glen's flow law). Uncertainty in the temperature directly translates to uncertainty in the inferred damage (Borstad et

al., 2013), as the inversion relies on determining a limiting value of strain rate that you can expect for ice at a given temperature. Therefore, you need to describe how you initialized the value of 'B' for the inversion (did you use a temperature field? Or a uniform initial 'B'?).

In fact, it is not clear which of two different methods for inverting for damage you have chosen. Borstad et al. (2012) inverted for D directly, whereas Borstad et al. (2013) calculate D as a post-processing routine after inverting for the rate factor 'B'. The latter is a more robust method of determining damage (and backstress, which the authors mention briefly without describing how this is calculated). I have no idea how you calculated damage, or how much confidence to have in your results since you didn't provide any details.

Inverting for basal friction and ice rheology involve a lot of assumptions and parameters, which are characteristics you used to criticize damage mechanics (all models involve assumptions and parameters). How did you initialize your inversion? Inversion results depend on reasonable initial guesses, so how did you initialize the basal friction? And the flow rate factor B? And D_o if you inverted for damage directly?

Did you use regularization to penalize sharp gradients in the inversions? If so, how did you determine the appropriate level of regularization? What metrics of goodness-of-fit did you look at to measure the goodness of fit between the modeled and observed velocities? Such metrics should be reported for comparing to other studies, otherwise the reader has no idea whether your initialized model, upon which your entire analysis is based on, is any good.

From lines 27-20 it seems that you have both a melting rate and a thickening rate on floating ice? How does this work? Where does this come in to your analysis or calculations?

Figures
Figures are hard to read, the color scheme is rather difficult to interpret (and I'm not color blind, but the figures are certainly not color-blind friendly; choose a perceptually uniform color scale). The scale is too large to be able to interpret much detail in the difference between the damage and probabilistic approaches.

The grounding line needs to be shown in the figures.

There are many fractures visible in your figures that are not represented with the ticks that you have placed for "identified" fractures. I'm not sure I have confidence in your "observations" against which you are comparing the different models.

Line-by-line comments:

P1, L9: 50% improvement in what specifically?

P1, L10-12: what kind of grouping? What insight is gleaned from this?

P1, L21: not sure the reference supports the assertion that ice shelf calving is the biggest source of uncertainty in sea level rise estimates

P2, L2: not sure this reference supports the claim about increased calving from 1998-2003 and concerns about ice sheet stability

P2, L3: increased with respect to what? some reference amount or time?

P2, L6: again, what do you mean by "increeased" calving?

P2, L17: the correct term is "Linear Elastic Fracture Mechanics"

P2, L22: you need to be more specific when criticizing previous work: approximation of what? how is it "only" first order? describe what this actually is.

Equation 1: no need to show this equation if you are just describing it among other methods of representing calving. Either show equations for all methods, or none.

P2, L28 (and L31, and elsewhere): avoid staring sentences with "Also, ..."

Interactive
comment

P2, L30: this is important background material, especially since you are critical of damage mechanics and use it to compare with your new method. The table is helpful as a reference, but you should describe the different approaches in the text here. Describe what "limitations" and "uncertainties" are associated with these methods, especially since you state that these are reasons to doubt them.

P3, L2-3: which is it? where they are located or where they initiated?

P3, L14: damage mechanics is not limited to "small" fractures

P3, L14-15: this is incorrect and misleading. Damage mechanics is much more general. You might be warranted in criticizing one particular study for its limitations here, but you cannot cast doubt on all of damage mechanics in this way.

P3, L18: "very high frequency of fracturing" needs a reference or better description here, this is quite vague.

P3, L18-19: this is the whole point of using inverse methods to infer the location of damaged ice, as you don't need to "see" the fractures from visible imagery, rather you infer them from their influence on velocities/strain rates

P3, L21-11: LEFM isn't about fracture propagation actually, people just assume that fractures will propagate until the stress intensity factor falls below some value, but this is not actual fracture modeling.

P3, L22-24: this is too vague of language for a scientific paper. You need to describe the crevasse depth criterion rather than just name it in quotes. You have to say what you mean by "interesting" results, and describe the "different mechanics" that preclude its use for ice shelves.

P3, L28: damage mechanics is about both the initiation and subsequent evolution of fractures.

P3, L31: Linear Elastic Fracture Mechanics

P3, L32: what do you mean that calving occurs "mainly" at the ice shelf front?

P4, L1-10: Borstad et al (2016) showed how to model damage evolution in an ice shelf based on stresses. Although this didn't explicitly treat calving, it provides a foundation for calving using damage evolution.

P4, L12-13: unjustified statement. This method has not been applied to ice shelves, so you have to support your claim that this can help understand calving in Antarctica "much better"

P4, L13-15: not true! look at Borstad et al. (2016) for modeling ice shelf damage. Also, this is not really true in general, a properly parameterized damage model doesn't need to "know" where fractured zones are in advance

P4, L18-20: unsubstantiated claim, damage mechanics has been successful in modeling very complex fracture processes in a wide range of engineering and natural materials... this is a disingenuous characterization of damage mechanics to suit your needs here

P4, L21-22: all methods have uncertainties, and this is not a reason to criticize them. You have to be a lot more specific.

P4, L25: see Borstad et al (2016)

P4, L26: this is value language and sounds unscientific. You need to be much more specific when criticizing other models/papers. All models have assumptions and uncertainties, and this is especially the case for some of the decisions you have made about setting up your statistical model.

P4 L29-30: this is only the case when inverting for damage, not in general damage evolution or when using the method described in Borstad et al. (2013) for calculating damage as a post-processing step after an inversion for the rate factor 'B'. It's not actually clear which you did, and the latter does not require selecting an initial damage.

P6 L20: it's unclear how and why you used the components of the principal stress axes, which are components of a unit normal vector describing the direction of the principal stresses. As this is a 2-component vector, how did you use these components? Individually, or combined somehow? What does a unit vector have to do with fracturing?

P7 L15: this should be strain rate (with a dot)

P7 L16: where did the strains come from? Why did you use strains?

P7 L19-20: not true: the axis defines the unit normal in the direction of the stress. The sign of the stress determines whether it is compressive or tensile.

P8 L1: what is a "vertical bend"?

Equation 9: of course, this would be implicitly accounted for by increased stress/strain rates in a properly formulated physical model.

P8 L12: completely arbitrary, it seems. And you criticized damage mechanics for having uncertainties and arbitrary factors?!

P8 L13-16: you need to show this in the results. Has this been found before? What might be the physical reason for such a grouping of fractures? What time range of satellite data did you look at, e.g. is this a robust finding in space and time?

P9 L25-26: but this says nothing about the depth of the fracture, so an insignificant surface crevasse will be represented the same as a through-thickness rift?

P10, L4-5: this seems like a needlessly simplistic definition, and avoids using the knowledge we have about fracturing: fractures are a response to high stresses! You cannot say that some node has a 50-50 chance of being fractured if you have modeled ice stresses but have just chosen to ignore them.

P18 L23-25: not true, these are geometric factors that influence the stresses, which can be resolved in a properly-formulated physical model

[Figure]

P20 L13: this is not true, a lot of fracture mechanics pre-dates this work

References

Borstad, C., A. Khazendar, B. Scheuchl, M. Morlighem, E. Larour, and E. Rignot (2016), A constitutive framework for predicting weakening and reduced buttressing of ice shelves based on observations of the progressive deterioration of the remnant Larsen B Ice Shelf, Geophys. Res. Lett., 43(5), 2027–2035, doi:10.1002/2015GL067365.

Colgan, W., H. Rajaram, W. Abdalati, C. Mccutchan, R. Mottram, M. Moussavi, and S. Grigsby (2016), Glacier Crevasses: Observations, Models and Mass Balance Implications, Rev. Geophys., 54(1), 119–161, doi:10.1002/2015RG000504.

Fürst, J. J., Durand, G., Gillet-Chaulet, F., Merino, N., Tavard, L., Mouginot, J., Gourmelen, N., and Gagliardini, O. (2015), Assimilation of Antarctic velocity observations provides evidence for uncharted pinning points, The Cryosphere, 9, 1427-1443, https://doi.org/10.5194/tc-9-1427-2015.

Matsuoka, K. et al. (2015), Antarctic ice rises and rumples: Their properties and significance for ice-sheet dynamics and evolution, Earth Sci. Rev., 150, 724–745, doi:10.1016/j.earscirev.2015.09.004.

---

## Author Comment (AC2) · 24 Aug 2017

We would like to thank the referee for his/her feedback. Please, find our response below.

RC: "This is not serious text - nothing breaks because it is close to mountains or a grounding line. Such parameters may be correlated to fracture only if they correlate indirectly via ice mechanics. This is like correlating drowning accidents to ice cream eating."

We do not claim that fracturing is directly influenced by grounding lines or mountains. However, statistically, the occurrence of those events correlate and there is a physical basis that explains this correlation. We explain this in greater detail in the revised ver-

sion of the manuscript. It is absolutely true that correlation does not mean causation, but more crevasses are observed close to the grounding line due to tidal deformation, or close to nunataks because of high lateral drag. These two physical processes are generally not included in ice sheet models and our model provides a simple parameterization for crevasse propagation in these regions based on a comprehensive analysis. Here we are using a statistical approach and so the method is not "directly" based on ice physics. In addition, it is important to note that we do not explain the formation of fractures only by proximity to mountains. We include all the parameters that directly can cause fracture formation such as stresses and viscosity.

RC: "It is rather easy to construct such measures, but there is nothing to be learned from them. The connection are superficial. For this paper to be published a more relevant crevasse formation measure should be constructed. All included parameters should at least be thoroughly motivated"

We observed over 35 glaciers and thousands of fractures, our results are not based on a few occurrences of fracturing being correlated to our selected observation parameters. Our proposed approach is used to parameterize fracture formation processes and not to invent a new physical model of fracturing.

RC: "For this paper to be published a more relevant crevasse formation measure should be constructed. All included parameters should at least be thoroughly motivated"

Based on your comments we change the mentioned sentences to be clearer: Term "mountains" changed to "edges of glaciers/ice shelves". Here we describe why each parameter that we include in our analysis is indeed known to have an impact on crack propagation (and this is added to the new version of the manuscript):

Back stress: Back stress provides an additional compressive stress resisting forward motion of glacier ice. [1]

Effective strain rate and Principal strain rates: if the strain rate is sufficiently high,

crevasses can propagate to greater depth [2]. In addition, stresses can trigger brittle fracturing but, to model a gradual vicious process, strains have to be taken into account.

Horizontal strain gradient: Fractures can be formed in zones where strain experience significant variations.

Principal stress axes: We removed this parameter from the revised version. We agree that there was a confusion using the principal axes. However, this does not affect the results due to the fact that this parameter has a low impact on the most of glaciers/ice shelves (smaller than 0.1 correlation).

Principal stress: the sign of the principal stress determines whether it is compressive or tensile.

There are a number of parameters such as velocity, surface slope and a curvature of a glacier channel that are included in the calculation of the stress field, but for our method, we look at each component separately:

Bed and surface slope: On a steeper slope, shear stress increases and can lead to fracturing (e.g. ice fall is an extreme case).

Surface gradient change: If there is a sudden change in a surface elevation the stress can increase causing fracturing.

Curvature: Fractures can be formed when a glacier flows over a horizontal bend.

Friction coefficient: low friction will lead to a higher sensitivity to membrane stresses, which can lead to more crevassing in tensile mode.

Rheology B: Stiffness of ice that can affect fracturing.

Thickness: Is included due to its physical relation to fracture mechanics

Proximity to the ice front: Included into the analysis due to observations of satellite images where a lot of fracturing occurs near ice shelf terminus.

Proximity to the grounding line: This parameter is included because tidal deformation at the grounding line can cause fracture formation, as observed in satellite images.

Proximity to edges of a glacier: What we meant was the edges cause lateral drag and fracturing. Thus, proximity to mountains (edges) ended up being predictive because lateral friction along the edges of glaciers is not generally considered in ice sheet models when stresses are calculated, therefore the stress field alone does not have the full ability to predict zones of fracture formation. Without the lateral drag only transverse, longitudinal and radial splaying crevasses can be predicted. They are all formed due to opening stress and are normally considered in former methods. However, the prediction of marginal crevasses requires a parameterization of the lateral drag.

Added to P8L14:

Generally, the lateral friction along the edge of a glacier is not considered in ice sheet models when stress is calculated. Therefore, the stress field alone does not have the full ability to predict zones of fracture formation, because without the lateral drag only transverse, longitudinal and radial splaying crevasses can be predicted. They are all formed due to opening stress and are normally considered in existing damage modelling methods. However, the prediction of marginal crevasses requires a parameterization of the lateral drag.

Added:

"The formation of fractures is a complex process that has not been effectively parameterized in ice sheet models to be applied to any glacier in Antarctica. "Previous analysis based on damage accounts for stresses, thickness and viscosity. However, for statistical analysis other factors can be equally important (such as proximity to edges of a glacier and the grounding line as well as the curvature of a channel), which might be crucial for modelling of the fracture formation in Antarctica."

References:

[1] Kenneally, James P., and Terence J. Hughes. "Fracture and back stress along the Byrd Glacier flowband on the Ross Ice Shelf." Antarctic Science16.3 (2004): 345-354.

[2] Benn, Douglas, and David JA Evans. Glaciers and glaciation. Routledge, 2014.

---

## Author Comment (AC3) · 24 Aug 2017

We would like to thank the reviewer for his careful and detailed comments; it helped us significantly improve several aspects of the manuscript that were not clear, such as the background description and the details about the model setup. However, we think that there have been some misunderstandings about the point of our paper, which might be due to the language and the writing style we used. We did not intend to prove that damage mechanics should not be used in ice sheet models and, certainly, it was not to claim that the whole concept of damage mechanics is incorrect. Our paper is meant to provide an alternative way for improvement of fracture modelling in Antarctica, which can be used in parallel with physics-based models.

Our approach is to try to develop a model that can predict the location of fractures in glaciers and ice shelves in a low-advection cycle. We do this by building a statisticsbased model that includes a combination of predictor factors. Our selection of these factors was based on a careful literature search of what other authors considered to affect fracturing and calving. The reviewer is correct that some of them turned out to be irrelevant, but we chose to mention them in our paper in order to provide full information about our results. Furthermore, fractures occur in ice at particular geographical locations for reasons that can be explained by physical processes (which is the basis of the damage approach). Once a fracture has formed, it is transported downstream by the flow of the glacier. Therefore, any physical process-based model should predict zones of fracturing that are upstream of the observed fractures. There are cases in Antarctica where fractures are visible in satellite imagery yet damage models do not identify zones of damage upstream (see enclosed Fig. 1, Fig. 2, Fig.3). It is this issue that our manuscript seeks to address, by trying to use a logistic regression analysis together with satellite images.

The reviewer raised a valid issue that we did not include the advection and that the locations where fractures are visible are not necessarily the locations where they were formed. In this research, we did not focus on the damage advection nor the probability advection because the focus was on modelling the location of fractures and not on describing if they were advected to this location or directly formed there. For example, one might argue that by claiming that we correctly predicted 3 fractures it is possible that we predicted only one of them and the other two fractures were visible on the satellite image just due to their advection downstream. However, the main point we introduce here is that if the stress was becoming compressive (or other factors we include) the fractures would close and would not be visible on the satellite image. The fact that we see fractures provides the basis to think that there are certain factors that act in favour of this fracture continuing to exist, irrespective of whether it forms in this particular location or remains open after being advected from upstream. In this

TCD
**fractures.**

We did not provide a detailed description of damage mechanics because it was not the focus of this paper. Instead, we focused more on the details of our method. We utilized the existing damage method that was already implemented in ISSM and, therefore, did not modify or add anything that needed a detailed description beyond what we referenced in existing published articles. We have taken the reviewer's feedback into account and have added the initialization of the damage method we used in the revised manuscript. Overall, there are many valid issues with the wording and the way we explained our method, but we did not identify a strong argument in the review comments that our approach does not make a valuable contribution to the existing models. We intend to add all the suggested modifications and rewrite the manuscript to clarify the above points, as well as the details on the model setup.

Respond to the line-by-line comments:

RC: "The grounding line needs to be shown in the figures"

Corrected.

RC: "There are many fractures visible in your figures that are not represented with the ticks that you have placed for "identified" fractures. I'm not sure I have confidence in your "observations" against which you are comparing the different models."

For the plotting purposes, we did not mark each single fracture but some of them a few in order to show the regions where fractures were observed.

RC: This manuscript describes a new probabilistic method for representing the location of fractures in Antarctic ice shelves (although the title indicates "glaciers" it is really ice shelves that are the focus).

Changed the title to: A statistical fracture model for Antarctic ice shelves. RC: "The introduction and background is rather meandering, and paints a quite critical picture of the use of fracture mechanics and damage mechanics for representing fractures in
glaciers."

First, we do not intend to criticize linear elastic fracture mechanics in our paper. Moreover, whereas we talk about uncertainties and shortcomings of the damage-based method in ice sheet models we do not claim that it should not be used. In fact, we said: "...we can see that to date Continuum Damage Mechanics [7] and Linear Fracture Mechanics [11] are the most common methods to model fracture formation. In addition, Krug et al. (2014) [8] showed that combining these two methods provides a significant improvement towards modelling of calving."

We did not intend to draw a critical picture of damage mechanics, but rather demonstrate that it has some deficiencies when applied to Antarctic ice shelves in order to motivate our approach. We did not claim that the damage method should not be used at all. As the reviewer mentioned, every method has uncertainties and we suppose that damage-based method is not perfect either. Therefore, we suggest another way of predicting fracture initiation. We present our method as an alternative and we use damage mechanics to compare our results to existing approaches.

To make it clear we add:

P3L5: "It is important to note that the physics-based methods such as LEFM and damage mechanics are necessary when modelling fractures in Antarctica. In this study, we do not intend to substitute them but rather find a method that can improve on some aspects and cases when physics-based models cannot predict fracture formation."

RC: "Indeed, the naive reader might be left with the impression that these methods are completely arbitrary and without merit."

Our main focus was to describe the motivation for developing an alternative method based on observations and statistics. We have added a better description of damage mechanics in our manuscript and articulate better the advantages of the existing methods.
Changed: P4L17 "Damage is a parameter that can be used to determine failure of ice..." Added: P4L18: Damage mechanics has been successfully applied in a large number of engineering research and to model damage at individually selected ice shelves such as the Ross, Filchner– Ronne, Amery [1], Larsen C [3] and Larsen B ice shelves [4].

RC: "It does not appear that the authors have a very thorough awareness and understanding of the damage mechanics literature as it has been applied in recent years to glacier and ice shelves. Inversions for damage of the type presented in the manuscript only produce damage in areas where fractures are actively forming or widening. It is important to distinguish between the "high-advection lifecycle" and "low-advection lifecycle" crevasse definitions of Colgan et al. (2016)."

We outlined the main aspects of damage mechanics following Krug et al., 2014 [8]. We appreciate the reviewer pointing out one of the recent papers we were not aware of. We have added the method described by Borstad et al. (2016) to the revised version of the manuscript. High-advection and low-advection life cycles described in Colgan et al. (2016) is worth mentioning, however, it would not affect the results of our method as it accounts for both.

Added: P6L8: Fractures in a certain location can be a result of advection of fractures from upstream (high-advection) or a result of the local stresses (low advection) (Coglan et al.,2016). It is important to note that in this method we do not distinguish between these high-advection and low-advection cycles. The main goal of the proposed method is to determine the most likely location of fractures without focusing on their initial source.

RC: "If crevasses have advected far from where they formed, then the appearance of a crevasse is not a representation of local stresses/strain rates! In the presented statistical method, no difference is made between where fractures are initiating versus where they have been advected for long distances after initiation. This history-dependence is
very important and is a key shortcoming of the present approach.

The reviewer is correct that the appearance of crevasses is not always a representation of the local stress (though in a low-advection cycle it is). However, the appearance of crevasses in a certain region reflects the fact that the stress is not compressive, which keeps them open even if they were advected from upstream and not directly formed in the region. We did not articulate this point well enough in our first submission, but make it clear in our revised version.

We understand that where fractures are formed is not the same as where they are advected, but the focus of this paper is on knowing where the fractures are located (without an attempt to determine where exactly they were formed as the main purpose of this is to predict the calving front location), the propagation and advection of fractures is out of scope of this research.

Added: P6L8: It is important to note that where fractures are formed is not necessary where they are located.

Added: P6L10: The main point we introduce here is that even if a fracture did not form at a particular location, but advected there with the ice flow when a certain combination of factors is met the fractures would close and would not be visible on the satellite image. The fact that fractures can be seen provides an indication that there are certain factors that act in favour of the fractures to exist, whether they formed in a particular location or remain open after being advected from upstream. Using this approach we do not directly model advection but predict the locations of both initiated and advected fractures.

RC: It also makes the comparison with the damage inversion a sort of apples-vsoranges comparison. The two approaches shouldn't necessarily produce the same thing." We compare each of the methods with observations. The damage-based method based on inversion should only predict the location of the formation of fractures only (unless it's a low-advection cycle). Therefore, this approach should predict TCD
damage zones at the formation location or upstream from observed fractures, which is not always the case in our damage inversions.

To clarify this as well as the damage initialization we added a new section 2.3 "Damageinversion set up" where we describe the model details and initialization. We added: "We utilize the damage-based model in order to compare it with observations and to identify areas where it can and cannot predict observed fractures. Thus we do not compare our probability-based model with the damage model directly; rather, we evaluate the ability of both to predict the formation of fractures in ice."

RC: "As for the statistical approach, there are many arbitrary and strange choices in its formulation. It seems as if every possible observational factor has been thrown into the mix just to see what comes out. Surprisingly, factors like the principal stress axes (components of unit normal vectors) and proximity to mountains end up being predictive, even if they have no physical relation to fracture mechanics, which should be the foundation of even a statistics-based fracture model."

We tested a number of observations to see if they had any effect on the results. The observational factors were not "thrown into the mix", but chosen from a list of factors considered by other authors, analyzed to qualify the effect that they had on fracturing and removed if found not to be relevant/required. We now clarify the choice of the parameters more clearly.

The term "mountains" here is indeed misleading. What we meant was proximity to the edges that can cause lateral drag and fracturing. Thus, proximity to mountains (edges) ended up being predictive because lateral friction along mountains is not generally considered in ice sheet models when stresses are calculated, therefore the stress field alone does not have the full ability to predict zones of fracture formation. Proximity to mountains is not a physical parameter here, but a parameter that is used for parameterization. Without the lateral drag only transverse, longitudinal and radial splaying crevasses can be predicted. They are all formed due to opening stress and are nor-
mally considered in former methods. However, the prediction of marginal crevasses requires a parameterization of the lateral drag. We now explain this better in our revised manuscript. The term "mountains" is replaced with the term "edges".

Added to P8L14: Generally, the lateral friction along the edge of a glacier is not considered in ice sheet models when stress is calculated. Therefore the stress field alone does not have the full ability to predict zones of fracture formation because without the lateral drag only transverse, longitudinal and radial splaying crevasses can be predicted. They are all formed due to opening stress and are normally considered in existing damage modelling methods. However, the prediction of marginal crevasses requires a parameterization of the lateral drag.

RC: "In the end, the most heavily influential factors in the statistical model are factors that would lead to higher stresses (and thus higher damage or predictions of crevasse depth) in a properly formulated and initialized model. This would seem to actually argue in favour of continuing with physical models such as the continuum damage mechanics models that are roundly criticized in the manuscript."

While a perfect model would capture all of the factors, at this stage ice sheet models do not include important factors such as tidal deformation at the grounding line or lateral friction along the edges of glaciers. Thus, in our method we suggest a way to parameterize those factors in order to predict zones of fracture initiation. We do not argue that damage should be abandoned. All methods are valuable when combined and they can improve our limited knowledge about fracturing in Antarctica.

RC: "The manuscript would benefit from a careful rewrite to avoid the use of vague or unscientific language, and to better describe the background material and theory. The modelling methods and results need to be described and shown in much more detail in order for a competent peer to be able to attempt to reproduce the study and check its findings." We have added all the details in the revised version.

RC: "The description here of the use of damage mechanics is lacking in sufficient de-

TCD
tail both to judge and to be able to reproduce the study. The authors are very vague in describing how they calculated damage for comparing to their statistical results. I am suspicious of the damage results, although there is simply not enough information for me to say one way or another whether their results are reasonable. I am not convinced that the new statistical model is actually better, in any quantifiable sense, than a properly formulated damage model."

We describe our calculation of damage as an inversion that was already implemented in ISSM, based on inversions suggested in [2] and [4]. We were focused on describing all the details of our probabilistic method because we only used the damage inversion as a way to compare our approach to existing methods.

We have added more details on the damage inversion that was used in the revised version of the manuscript.

We show the results of both methods and their ability to predict fracture formation zones. It is difficult to always distinguish on satellite images where fractures are formed and where they are advected, but we believe that, in general, our model was able to predict those regions well. The proposed method has uncertainties, but overall it describes the observations well.

RC: P1, L9: 50% improvement in what specifically?

Added: improvement in predicting the location of zones of fractures.

RC: P1, L10-12: what kind of grouping? What insight is gleaned from this?

Changed to: We found that Antarctic ice shelves can be classified into groups based on the factors that control fracture location. It gives insights on the fact that factors that trigger fracturing as well as sustain existing fractures advected from upstream can vary from one ice shelf to another.

RC: P1, L21: not sure the reference supports the assertion that ice shelf calving is the biggest source of uncertainty in sea level rise estimates
Changed to: with a great uncertainty related to calving dynamics, as it is not a part of the CMIP5 simulations (Church et al., 2013) [5].

RC: P2, L2: not sure this reference supports the claim about increased calving from 1998

Corrected to: Shepherd et al., 2012 [10]

RC: P2, L3: increased with respect to what? some reference amount or time?

Added: increased number of calving events in the last two decades. For example, break off Larsen B and Larsen C ice shelves.

RC: P2, L6: again, what do you mean by "increased" calving?

Changed to: "..showed that this increased the number of calving events."

RC: P2, L17: the correct term is "Linear Elastic Fracture Mechanics"

Corrected.

RC: P2, L22: you need to be more specific when criticizing previous work: approximation of what? how is it "only" first order? describe what this actually is.

Added: first order approximation of a calving rate that includes spreading rates of the first order and assigns all higher order terms to zero. The idea is based on the observations of the increase of calving rate with along-flow ice shelf spreading rates and the spreading rates perpendicular to the calving front.

RC: Equation 1: no need to show this equation if you are just describing it among other methods of representing calving. Either show equations for all methods, or none.

Removed.

RC: P2, L28 (and L31, and elsewhere): avoid starting sentences with "Also, ..."

Removed in many places throughout the text.

**TCD**
RC: P2, L30: this is important background material, especially since you are critical of damage mechanics and use it to compare with your new method. The table is helpful as a reference, but you should describe the different approaches in the text here. Describe what "limitations" and "uncertainties" are associated with these methods, especially since you state that these are reasons to doubt them. We have presented the main limitations and uncertainties of these studies in section 2.1: "Current state of calving computations in ice sheet models". To clarify that we added: In addition, Krug et al. (2014) showed that combining these two methods provides a significant improvement towards modelling of calving (the uncertainties and limitations of these methods are described in section 2.1)

RC: P3, L2-3: which is it? where they are located or where they initiated? Where they are initiated. Changed to: it is essential to know where those fractures are initiated.

RC: P3, L14: damage mechanics is not limited to "small" fractures Removed small.

RC: P3, L14-15: this is incorrect and misleading. Damage mechanics is much more general. You might be warranted in criticizing one particular study for its limitations here, but you cannot cast doubt on all of damage mechanics in this way.

Changed to: The damage-based method is used in ice sheet models to calculate zones where ice is weakened due to a formation of fractures.

RC: P3, L18: "very high frequency of fracturing" needs a reference or better description here, this is quite vague.

Changed to: as well as a high frequency of fracturing as can be visible on satellite images (up to every 50 metres).

RC: P3, L18-19: this is the whole point of using inverse methods to infer the location of damaged ice, as you don't need to "see" the fractures from visible imagery, rather you infer them from their influence on velocities/strain rates

Added: Satellite observations alone are difficult to use because of the very high reso-
lution required to see all the small fractures and snow can often cover the fractures in the image; in which case, inverse methods are often used [3,4]

RC: P3, L21-11: LEFM isn't about fracture propagation actually, people just assume that fractures will propagate until the stress intensity factor falls below some value, but this is not actual fracture modelling.

Corrected: The LEFM approach in ice sheet models is focused on calculating a stress intensity factor around fractures and assuming that they propagate until falls below a certain critical value.

RC: P3, L22-24: this is too vague of language for a scientific paper. You need to describe the crevasse depth criterion rather than just name it in quotes. You have to say what you mean by "interesting" results, and describe the "different mechanics" that preclude its use for ice shelves.

Modified: Water-depth calving models second uses flotation criteria to estimate the location of glacier terminus. It allows linking calving to glacier dynamics as well as surface melting when applied to marine terminated glaciers in Greenland (Nick et al., 2010). However, it cannot describe calving at the floating ice shelves in Antarctica due to the fact that the water-depth requires glacier to calve once it reaches flotation.

RC: P3, L28: damage mechanics is about both the initiation and subsequent evolution of fractures.

Added: The evolution of damage can be also modelled using approach suggested by Borstad et al., 2016. There are a number of other studies suggested [Krug et al., 2014; Albrecht and Levermann, 2014, Pralong and Funk, 2005; Duddu and Waisman, 2012), but they might be not applicable in a generalized large scale case. RC: P3, L31: Linear Elastic Fracture Mechanics

Corrected.

RC: P3, L32: what do you mean that calving occurs "mainly" at the ice shelf front?

TCD
Corrected: occurs at the ice front.

RC: P4, L1-10: Borstad et al (2016) showed how to model damage evolution in an ice shelf based on stresses. Although this didn't explicitly treat calving, it provides a foundation for calving using damage evolution.

Added: Finally, Borstad et al (2016) showed that evolution of damage in an ice shelf could be modelled using the stress field. This method has a number of advantages as it allows calculating mechanical ice weakening and predicting the degradation of ice shelves. The constant parameters that define damage have not been tested for other ice shelves apart from Larsen B, and so it is not demonstrated whether the approach is valid for other locations and settings. The validity of the parameter values can only be tested when an ice shelf undergoes pronounced mechanical changes, as did the Larsen B ice shelf. Moreover, the error in the observed ice temperature can be crucial in affecting the accuracy of the viscosity parameter (Bassis and Ma, 2015) [1].

RC: P4, L12-13: unjustified statement. This method has not been applied to ice shelves, so you have to support your claim that this can help understand calving in Antarctica "much better"

Modified: We present the main studies of calving in Table 1 and among them a proposed parameterization of calving by Krug et al. (2014) that is based on combining damage and fracture mechanics. This method is more complex in comparison to the other approaches suggested before 2014 as it allows for both viscous and elastic behaviour and it is able to reproduce development of small crevasses over a long time scale. However, this method has not been applied to ice shelves in Antarctica.

RC: P4, L13-15: not true! look at Borstad et al. (2016) for modelling ice shelf damage. Also, this is not really true in general, a properly parameterized damage model doesn't need to "know" where fractured zones are in advance

Corrected: To apply the method suggested by Krug et al. (2014) to any ice shelf, the
modelled fractured zones need to be in a good agreement with the observed surface fractures. Therefore, modelling of the formation of the fractured zones is an important basis for the consequent estimation of fracture depth as well as calving and it needs to be described in the ice sheet models in an accurate way. On the other hand, the method proposed by Borstad et al., 2016 does not require modelling of the fracture formation zones, but it can only account for the viscous propagation of damage. RC: P4, L18-20: unsubstantiated claim, damage mechanics has been successful in modelling very complex fracture processes in a wide range of engineering and natural materials... this is a disingenuous characterization of damage mechanics to suit your needs here

We indeed were not looking at application outside of glaciology. We are proposing an alternative approach based on statistics. We do not need the other methods to be "wrong" in order to suggest an alternative approach and revised the text accordingly. To clarify this we add: This method has been successfully applied in a wide range of engineering problems, but has not yet been applied at the scale of the Antarctic continent with good agreement with observations of actual rifts and crevasses. It is possible that the nature of fracturing of the ice sheet is more complex and requires a combination of methods to better parameterize fracture formation. In fact, it is claimed that damage may not be justified for glacier ice, especially at the ice front [9].

RC: P4, L21-22: all methods have uncertainties, and this is not a reason to criticize them. You have to be a lot more specific.

Changed: The propagation of damage is usually calculated using an advection equation and a source function (Krug et al., 2014, Albrecht and Levermann, 2014). However, this method has a number of uncertainties such as the choice of the source function as well as the number of decisive parameters that define the damage evolution (damage threshold, initiation threshold and the enhancement factor).

RC: P4, L25: see Borstad et al (2016)

Modified: In fact, the source function is the controlling factor in the damage propagation
and the hypothesis of what should be used as the source function has been proposed by Pralong and Funk, 2005 as well as Krug et al., 2014, but they are not generalized to be applicable to all ice shelves. Recently Borstad et al. (2016) suggested a framework where instead of computing a damage source term, as is usually done, the damage is part of a generalized constitutive relationship. This can significantly improve the accuracy of the zones where the ice is weakened by illuminating the uncertainties related to the source function. However, the used value of stress threshold 130 +/-14 kPa found for Larsen B ice shelf may vary for other ice shelves. In addition, when performing an inversion the greatest uncertainty comes from the sparse data about ice temperature. Moreover, it accounts only for the vicious behaviour of fracture formation and ignores the elastic behaviour.

RC: P4, L26: this is value language and sounds unscientific. You need to be much more specific when criticizing other models/papers. All models have assumptions and uncertainties, and this is especially the case for some of the decisions you have made about setting up your statistical model.

**Removed.**

RC: P4 L29-30: this is only the case when inverting for damage, not in general damage evolution or when using the method described in Borstad et al. (2013) for calculating damage as a post-processing step after an inversion for the rate factor 'B'. It's not actually clear which you did, and the latter does not require selecting an initial damage. In our case, we focus on the inversion, not the general case proposed by Borstad, 2016 [4], because we are interested in the zones of initiation of fracturing, not the propagation. The mentioned line refers to Borstad et al. (2012) [3], we did not say it was the case in any damage method.

Modified: this line is moved to the paragraph describing Krug et al. (2014) and Borstad et al. (2012)

RC: P6 L20: it's unclear how and why you used the components of the principal stress:

TCD
it's unclear how and, which are components of a unit normal vector describing the direction of the principal stresses. As this is a 2-component vector, how did you use these components? Individually, or combined somehow? What does a unit vector have to do with fracturing? We agree that using principal axes is misleading. We have removed this parameter from the list of predictors. This does not affect the outcome when modelling the location of fractures.

RC: P7 L15: this should be strain rate (with a dot)Í

Corrected.

RC: P7 L16: where did the strains come from? Why did you use strains?

Added: It is included because if the strain rate is high enough, crevasses can propagate at higher depth (Benn et al., 2014). In addition, stresses can trigger brittle fracturing, but to model a gradual vicious process, strains have to be taken into account.

RC: P7 L19-20: not true: the axis defines the unit normal in the direction of the stress. The sign of the stress determines whether it is compressive or tensile.

Corrected: In addition, principal stress shows the direction of the stress field, whether it is compressive or tensile (- or + sign).

RC: P8 L1: what is a "vertical bend"?

Added: vertical bend, which is an uplift of the ice surface.

RC: Equation 9: of course, this would be implicitly accounted for by increased stress/strain rates in a properly formulated physical model.

It is accounted for in ISSM. However, the idea here is to split all the factors into components, to see which ones play the more important roles.

RC: P8 L12: completely arbitrary, it seems. And you criticized damage mechanics for having uncertainties and arbitrary factors?! It is not arbitrary. We do not use this pa-

TCD
rameter in the model as it is. The parameter is a distance used to identify the curvature of the glacier; it does not affect the calculated curvature itself.

RC: P8 L13-16: you need to show this in the results. Has this been found before? What might be the physical reason for such a grouping of fractures? What time range of satellite data did you look at, e.g. is this a robust finding in space and time? The ice velocity increases when moving to the ice front, therefore the consequent increase in stress can lead to fracture formation. We provide a graph showing the relation between the number of fractures and the distance to the ice front in the revised version of the manuscript. We use satellite images from between 2011 and 2015. It was already included in the manuscript (P9 L9). There is not sufficient data available for each year for every glacier/ice shelf. However, for regions where high-resolution satellite images are available for every year, it is a robust finding and the location of crevasses is the same between the observed years.

RC: P9 L25-26: but this says nothing about the depth of the fracture, so an insignificant surface crevasse will be represented the same as a through-thickness rift? An insignificant surface crevasse will not be visible in the satellite imagery. The estimation of fracture depth is beyond the scope of this research. We do not include observations of rifts into the statistical framework, so there is no expectation at this stage that it would predict them. We did not add any additional information to the manuscript.

RC: P10, L4-5: this seems like a needlessly simplistic definition, and avoids using the knowledge we have about fracturing: fractures are a response to high stresses! You cannot say that some node has a 50-50 chance of being fractured if you have modelled ice stresses but have just chosen to ignore them. We do not ignore them and it is wrong to say that we have. We scale the probability field due to the fact that the statistical approach based on logistic regression requires scaling, as in general there are more non-fractured nodes than fractured. In addition, scaling is required for calculation of prior when applying Bayesian (it produces too large values when calculating it without the scaling due to the fact that there are a large number of non-fractured nodes (if the
probability of fracturing is less than 50 per cent))

RC: P18 L23-25: not true, these are geometric factors that influence the stresses, which can be resolved in a properly-formulated physical model. There is no such model at this point because currently ice sheet models do not include lateral drag along glacier/ice shelf edges and grounding line flexure. Creating such a model would be a research project in itself. Instead, we choose to approach this problem without changing the physical models. We suggest a method based on logistic regression as a way of parameterization.

RC: P20 L13: this is not true, a lot of fracture mechanics predates this work Changed: Most previous research on fracture formation has been focused on applying fracture mechanics as well as damage mechanics.

References:

[1] Bassis, Jeremy N., and Y. Ma. "Evolution of basal crevasses links ice shelf stability to ocean forcing." Earth and Planetary Science Letters 409 (2015): 203-211.

[2] Borstad, C. P., et al. "A damage mechanics assessment of the Larsen B ice shelf prior to collapse: Toward a physically-based calving law." Geophysical Research Letters 39.18 (2012).

[3] Borstad, C. P., et al. "Creep deformation and buttressing capacity of damaged ice shelves: theory and application to Larsen C ice shelf." The Cryosphere7.6 (2013).

[4] Borstad, Chris, et al. "A constitutive framework for predicting weakening and reduced buttressing of ice shelves based on observations of the progressive deterioration of the remnant Larsen B Ice Shelf." Geophysical Research Letters 43.5 (2016): 2027-2035. [5] Church, J.A., P.U. Clark, A. Cazenave, J.M. Gregory, S. Jevrejeva, A. Levermann, M.A. Merri eld, G.A. Milne, R.S. Nerem, P.D. Nunn, A.J. Payne, W.T. Pfeffer, D. Stammer and A.S. Unnikrishnan, 2013: Sea Level Change. In: Climate Change 2013: The Physical Science Basis. Contribution of Working Group I to the Fifth AssessTCD
ment Report of the Intergovernmental Panel on Climate Change [Stocker, T.F., D. Qin, G.-K. Plattner, M. Tignor, S.K. Allen, J. Boschung, A. Nauels, Y. Xia, V. Bex and P.M. Midgley (eds.)]. Cambridge University Press, Cambridge, United Kingdom and New York, NY, USA. [5] Colgan, William, et al. "Glacier crevasses: Observations, models, and mass balance implications." Reviews of Geophysics 54.1 (2016): 119-161.

[6] Douglas, B. and Evans, D.JA. Glaciers and glaciation. Routledge, 2014.

[7] (Kachanov, 1958) KACHANOV, Lazar M. "Time of the Rupture Process under Creep Conditions, Izy Akad." Nank SSR Otd Tech Nauk 8 (1958): 26-31.

[8] Krug, J., et al. "Combining damage and fracture mechanics to model calving." The Cryosphere 8.6 (2014): 2101-2117.

[9] Levermann, Anders, et al. "Kinematic first-order calving law implies potential for abrupt ice-shelf retreat." The Cryosphere 6.2 (2012): 273.

[10] Shepherd, Andrew, et al. "A reconciled estimate of ice-sheet mass balance." Science 338.6111 (2012): 1183-1189.

[11] Van der Veen, C. J. "Fracture mechanics approach to penetration of surface crevasses on glaciers." Cold Regions Science and Technology 27.1 (1998): 31-47.

TCD

---

## Author Comment (AC4) · 30 Aug 2017

RC: P3, L2-3: which is it? where they are located or where they initiated?

Where they are located.

Changed to: it is essential to know where those fractures are located.